# Predictors of binge drinking among adults in Autonomous Province of Vojvodina, Serbia: Data from the population-based behavioral risk-factors surveillance system

Tanja Tomašević[1,2], Ivana Radić[1,2]*, Vesna Mijatović Jovanović[1,2], Snežana Ukropina[1,2], Dragana Milijašević[1,2], Sonja Čanković[1,2], Radmila Petrović[3], Vladimir Petrović[1,2]

1 Faculty of Medicine, University of Novi Sad, Novi Sad, Serbia, 2 Institute of Public Health of Vojvodina, Novi Sad, Serbia, 3 University Clinical Center of Vojvodina, Novi Sad, Serbia

* ivana.radic@mf.uns.ac.rs

## Abstract

### Introduction

Alcohol use has been linked to harmful health outcomes. In this study, we examined the association between binge drinking and sociodemographic characteristics as well as health-related lifestyle factors among the adult population in Autonomous Province of Vojvodina (APV).

### Materials and methods

The data for this study were used from the newly established "Surveillance of Behavioural Risk Factors for Non-Communicable Diseases in Vojvodina" (SBRF-NCD-V) system in APV, Serbia. This cross-sectional study involved 3910 healthcare users aged 18 years and older, interviewed in 2024, across all 44 Primary Healthcare Centers in APV. The questionnaire used was adapted from the BRFSS instrument developed by the CDC. A multivariable binary logistic regression model analysed associations between binge drinking and sociodemographic, as well as health-related factors, stratified by gender to calculate odds ratios.

### Results

The prevalence of binge drinking in Vojvodina was high (19.3%). Higher odds of binge drinkers were recorded in males (OR=4.3; 95% CI=3.5–5.2; p<0.001), in younger categories compared to age groups 65 and over (p<0.001), never married females compared to married/living with a partner (OR=2.1; 95% CI=1.4–3.5; p=0.002), females with poor/very poor material status (OR=2.1; 95% CI=1.2–3.9; p=0.014), females who were former smokers (OR=1.8; 95% CI=1.1–3.3; p=0.033)

**Data availability statement:** The data used in this study are owned by the Institute of Public Health of Vojvodina. The Institute is responsible for data collection, management, and quality assurance, and guarantees the quality and integrity of the dataset. The authors have authorized access to and use of the data for this study. The dataset has been fully depersonalized. However, due to ethical and legal restrictions related to sensitive participant information, the data are not publicly available on the internet or in any public repository. We confirm that we have access to the data and can provide the de-identified dataset necessary to replicate the study findings in the form of an Excel database. However, access to the data by other researchers cannot be granted by the authors directly. Any requests for data access must be addressed to the Institute of Public Health of Vojvodina (email: izjzv@izjzv.org.rs), which is the data owner and will review and decide on all requests in accordance with ethical and legal requirements.

**Funding:** This study was supported by the Parliament of the Autonomous Province of Vojvodina, Serbia, according to the Decision of the Parliament of APV ("Official Gazette of the APV", No. 45/2023) "Program Task of the Special Public Health Program for the Territory of APV in 2024". The funding supported the survey implementation (data collection) and database processing only, and did not fund the research objectives, analysis, or the preparation of this manuscript.

**Competing interests:** The authors have declared that no competing interests exist.

or smokers (OR=2.8; 95% CI = 2.3–3.5; p < 0.001) compared to non-smokers, males who were not physically inactive (OR=1.7; 95% CI = 1.3–2.2; p < 0.001).

## Conclusion

Factors contributing to the high prevalence of binge drinking in APV, Serbia, were gender, younger age, never-married status in females, worse material status in females, former smoking in females, smoking for both genders, and physical activity in males. Introducing a continuous sub-national surveillance system could significantly improve the monitoring of factors associated with alcohol use prevalence. Identified sociodemographic factors could help health workers in primary care settings to screen and support patients at risk of, or engaged in risky alcohol use.

## Introduction

Alcohol has been widely used in many cultures for centuries, but it is associated with significant health risks and harms. Drinking alcohol is associated with risks of noncommunicable diseases such as different types of cancers, liver diseases, heart diseases, mental health, depression, anxiety, and alcohol use disorders [1]. Driving under the influence of alcohol (drink-driving) is a key risk factor for 27% of all road injuries, and road traffic crashes are the leading cause of death among people aged 15–29 years [2].

Risky consumption of alcohol means drinking above the recommended daily, weekly, or per-occasion amounts but not meeting criteria for alcohol use disorder [3]. The level of risk of alcohol-related harm depends on the amount consumed, frequency of drinking, the health status of the individual, age, gender, and other personal characteristics, as well as the context in which alcohol consumption occurs [1]. Binge drinking or high volume consumption over a short time, or excessive alcohol use, increases the risk of injuries, poisoning, violence [4], chronic disease, learning memory problems [5,6], increases the likelihood of unsafe sexual behavior and the risk of sexually transmitted infections and unintentional pregnancy, also increases the likelihood of a falls, burns, drownings, and car crashes [7]. Binge drinking is defined as consuming 4 or more drinks for women and 5 or more drinks for men on a single occasion [5,6].

According to research data on the burden of disease, alcohol consumption in Serbia is the eighth of the ten most common risk factors for premature mortality and disability [8].

In Serbia, drinking alcoholic beverages is a widely accepted social behavior, and alcohol use is deeply rooted in our society, and it is a part of most social rituals [9]. Although alcohol causes many health problems [1], data from the Serbian National Health Survey (SNHS) showed that approximately half of the population aged 20 and over consumed alcohol occasionally or daily, and that in the period from 2000 to 2019, there were no major changes in the frequency of drinking this substance. Of concern is the increasing number of adolescents who have engaged in binge drinking

in the previous 30 days in the reference period [10]. Despite this, Serbia does not have a formal national policy or National action plan for alcohol prevention [11].

In general population samples, alcohol use tends to vary along demographic, health, and social characteristics [12]. Factors influencing levels and patterns of alcohol consumption can be at the individual level and the societal level. Personal factors include age, gender, family circumstances, and socioeconomic status; societal factors include economic development, culture, social norms, and availability of alcohol [1]. Females are less often current drinkers than males, and when they drink, they consume less alcohol [11], therefore, contributing factors for alcohol use are male gender [13,14], smoking [13–15], being single, being young adults [14], and low level of education [13,16]. Moreover, in England, factors such as retirement, income, and marital status were also associated with high-risk alcohol consumption for people over 50 [17]. Evidence about the influence of socioeconomic status on drinking behavior is inconsistent in the literature [16,18,19]. Rates of alcohol use are higher for urban than rural residents, and rates of alcohol use disorder tend to be similar across rural and urban environments [20]. Alcohol is often consumed before, along with, or after other psychoactive substance use, and the comorbidity of alcohol and tobacco dependence is strong and well documented [11].

These patterns have been documented internationally, and we want to understand how they manifest in specific national contexts, which is essential for developing effective public health responses.

The Autonomous Province of Vojvodina (APV) is the northern Province within the Republic of Serbia, and daily alcohol consumption in Vojvodina is among the highest in Serbia [21]. To the best of our knowledge, there is little research in the literature about the prevalence and predictors of binge drinking in Serbia. Examining predictors of binge drinking could be used to target at-risk groups or to conceptualize intervention and prevention strategies for reducing risky alcohol use, which is a public health priority.

In this study, we examined the association between binge drinking and sociodemographic characteristics as well as health-related lifestyle factors among the adult population in Autonomous Province of Vojvodina.

## Methods

### Study design

This cross-sectional study was conducted by the Institute of Public Health of Vojvodina (IPHV) in 2024, from 20.05.2024. to 30.11.2024, as a part of the "Special Public Health Program for the Territory of APV: Surveillance of behavioral risk factors for non-communicable disease (NCD's) in the adult population in APV (SBRF-NCD-V)". The data were collected using a specially designed questionnaire based on the BRFSS instrument (Behavioral Risk Factor Surveillance System used by the American Centers for Disease Control and Prevention – CDC, United States of America) [22], which was translated into Serbian and administered to the study sample. The respondents were interviewed by trained health workers face-to-face using a mobile phone connected to a database on a computer program. The surveillance research system was established in 2023 by decision of the Parliament of APV, with the intention of being conducted annually. The paper presents the data from the second year of surveillance (2024).

### Sample size

The sample size consisted of 3,910 adults aged 18 and over who used healthcare at one of the 44 Primary Healthcare Centers (PHCs), across all 45 municipalities in APV. This population is covered by compulsory universal health care insurance at the rate of 98%, (a contract with general practitioners have 86.3% of insured persons), including the non-working population whose health insurance contributions are financed from the central state budget [23,24].

A representative sample of the adult population was created, and stratified according to gender, age, and type of settlement of 45 municipalities, based on the data of the Statistical Office of the Republic of Serbia on the number and structure of the population of APV (according to the Census data, total population of N = 1,740,230). A sample size was created

based on data on the prevalence of risk factors in this population, with a smoking prevalence of 35.5% being considered for the final sampling frame, with a precision of 1.5% and an accuracy of 95% [25]. According to the Census data, the number of surveyed individuals according to gender, age, and type of settlement (urban and rural/other) was previously planned by the IPHV (stratification). Each PHC received a separate stratification plan that included gender, settlement, and age for selecting participants in the sample. Exclusion criteria included persons with mental impairment and those who refused to be interviewed at any phase.

Ethical aspects of surveillance (compliance of protocol, questionnaire, written informed consent of respondents, and methodological instructions for implementation of the survey with participants' rights, ensuring that the study adheres to regulations, ethical guidelines, and regulatory standards) were approved by the Ethics Committee of the Institute of Public Health of Vojvodina (Decision No. 01–796/2–1 dated May 17, 2024). Each respondent signed the Informed Consent for participation in the survey, and it is stored separately from the database. The survey was anonymous, participation was voluntary, no incentives were provided, and the collected data were kept confidential on the IPHV server.

### Data collection, control, and data processing

Data were collected electronically, by healthcare workers who interviewed respondents using a mobile phone with personalized and authorized access to the software through a specially created software application accessed through the IPHV website. This approach enhanced data quality by minimizing data entry errors, validation checks, and automated skip patterns, which ensured that participants provided valid and consistent responses. The control of the collected data was carried out at several levels: supervisors, coordinators, and program managers. The primary database was transformed into a form suitable for processing in the statistical processing program SPSS (Statistical Package for Social Sciences), version 21.

### Outcome variable

**"Binge drinkers".** BRFSS defines binge drinking as "males having five or more drinks on one occasion, and females as having four or more drinks on one occasion" [5,6]. To discern alcohol consumption regarding binge drinking, the BRFSS survey first asks whether the respondent has consumed alcohol during the past 30 days, and if the answer is "yes," then there are inquiries about frequency with the following prompt: "One drink is equivalent to a 12-ounce beer, a 5-ounce glass of wine, or a drink with one shot of liquor". Further, the survey includes the following: "Considering all types of alcoholic beverages, how many times during the past 30 days did you have five or more drinks for men or four or more drinks for women on any occasion?" to determine whether the respondent had episodes in which they binged with alcohol. From these questions, BRFSS recodes data to produce a calculated variable determining whether the respondent had binge alcohol use in the past 30 days [26]. For estimating prevalence rates, the dependent variable was dichotomized into binge drinking (coded as 1) and non-consumption (coded as 0) in the last 30 days. Those who reported that they drink alcohol but did not report having 5 or more drinks for women or 4 or more drinks for men on one occasion were included in the category non-consumption.

### Explanatory variables

- Demographic and socioeconomic variables (factors included in the multivariable analysis): 1) Gender (male; female); 2) Age (18–34; 35–49; 50–64; 65 + years); 3) Place of settlement (urban; rural); 4) Marital status (married or living with a partner; divorced/separated; widowed; never married); 5) Education level (primary school or incomplete primary school; secondary school; university degree); 6) Employment status (employed or self-employed; unemployed; not in a labor force – those who are students, stay-at-home or unable to work; retired) [27]; 7) Self-assessment of material status (very good or good; average; poor or very poor).

Health-related and lifestyle risk factors variables (factors included in the multivariable analysis): 8) Smokers were considered respondents who smoked daily or occasionally; former smokers – were respondents who smoked at least 100 cigarettes during their lifetime but do not smoke anymore 9) Physical inactivity – physically inactive persons were those haven't been engaged in activity in the last month (excluding work) or engaged in activities such as running, muscle strengthening exercises, gardening or walking for exercise (yes; no).

## Statistical analysis

Data were analysed using SPSS (Statistical Package for Social Sciences), version 21, applying descriptive and inferential statistics methods. Categorical data were presented through frequency distributions and relative numbers. Differences between observed characteristics were tested using a $\chi^2$-test. Numerical data were presented through mean values and measures of variability. The associations of binge drinking with independent variables were examined by a multivariable binary logistic regression model. Missing values, as well as answers "I don't want to answer" or "I don't know" were excluded from the analysis. All values with $p < 0.05$ were considered statistically significant.

## Results

This cross-sectional survey included 3,910 persons, aged 18 and over, who were beneficiaries of primary health care, in APV, Serbia, in 2024. The average age of the respondents was 49.3 years (SD = 17.5). Among the respondents, the female population was 51.8%. Out of a total, 60.2% of respondents lived in urban settlements. According to marital status, more than half of the respondents stated that they live in marriage or live with their partner (57.3% of females and 61.7% of males). Out of the total, 19.7% of the respondents stated that they had never married, 15.3% were female, and 24.4% male. The largest percentage of residents has a secondary education (59.3%). According to the self-assessment of material status, the largest number of respondents indicated an average material status (46.2%). At the time of the survey, more than half of the respondents (61.6%) were employed or self-employed, while 32.6% were not in the labor force (27.0% of males and 37.8% of females). Smoking prevalence was 35.2%, and it was more prevalent among males than females (39.4% vs. 31.3%). There was 31.3% of the physically inactive population (29.2% of males and 33.3% of females). In APV in 2024, 45.8% of the population drank alcohol in the last 30 days (Table 1).

The prevalence of binge drinking in 2024 was 19.3% (95%CI 18.0–20.7). There were statistically significant differences in the percentage of binge drinkers among different sociodemographic groups aged 18 years and older. The highest prevalence was recorded in males (31.4%; 95%CI 29.1–33.7), the age 18–34 years (26.1%; 95%CI 23.1–29.3), in never-married persons (27.6%; 95%CI 24.3–31.2), and in the population with secondary education (20.4%; 95%CI 18.7–22.3). The highest percentage was also recorded in the employed population (23.8%; 95%CI 22.0–25.7), while the lowest was in the category not in the labor force (10.6%; 95%CI 8.9–12.5). The highest rate of binge drinkers was among smokers (31.1%; 95%CI 28.4–33.8), and among those who were not physically inactive (21.4%; 95%CI 19.7–23.1). In the respondents from rural settlements and among respondents with very good material status, the prevalence was the highest, but there was no statistical difference (Table 2).

Males were four times more likely to be binge drinkers (OR=4.3; 95% CI = 3.5–5.2; p < 0.001), and younger respondents had a higher chance of being binge drinkers than those who belonged to the category 65 and over (p < 0.001). The females respondents in the category never married were two times more likely to be binge drinkers (OR=2.1; 95% CI = 1.4–3.5; p = 0.002), than those who belong to the category married/living with a partner and females with poor and very poor material status had similar elevated odds compared to females with very good or good material status (OR=2.1; 95% CI = 1.2–3.9; p = 0.014). Higher odds of binge drinking were found in females who were former smokers (OR=1.8; 95% CI = 1.1–3.3; p = 0.033), while smokers had almost three times higher odds of being binge drinkers (OR=2.8; 95% CI = 2.3–3.5; p < 0.001), and considering gender- stratified odds ratios, this was relevant for

**Table 1. Sociodemographic and health-related lifestyle characteristics of respondents, Autonomous Province of Vojvodina (APV), Serbia, 2024 (n = 3,910).**

| Sociodemographic and health-related lifestyle characteristics | Total | Male | Female |
|---|---|---|---|
| | n (%) | n (%) | n (%) |
| **Gender** | | | |
| Male | 1886 (48.2) | 1886 (48.2) | |
| Female | 2024 (51.8) | | 2024 (51.8) |
| **Place of settlement** | | | |
| Urban | 2353 (60.2) | 1109 (58.8) | 1244 (61.5) |
| Rural | 1557 (39.8) | 777 (41.2) | 780 (38.5) |
| **Average age [years]** | $\overline{X}$ = 49.3 (SD = 17.5) | | |
| **Age groups [years]** | | | |
| 18-34 | 946 (24.2) | 487 (25.8) | 459 (22.7) |
| 35-49 | 1014 (25.9) | 521 (27.6) | 493 (24.4) |
| 50-64 | 1026 (26.2) | 477 (25.3) | 549 (27.1) |
| 65+ | 923 (23.6) | 400 (21.2) | 523 (25.8) |
| **Marital status** | | | |
| Married /Living with a partner | 2317 (59.4) | 1161 (61.7) | 1156 (57.3) |
| Divorced /Separated | 364 (9.3) | 152 (8.1) | 212 (10.5) |
| Widow/widower | 452 (11.6) | 109 (5.8) | 343 (17.0) |
| Never married | 768 (19.7) | 460 (24.4) | 308 (15.3) |
| **Education level** | | | |
| Incomplete primary school/primary school | 1127 (28.8) | 534 (28.3) | 593 (29.3) |
| Secondary school | 2318 (59.3) | 1165 (61.8) | 1153 (57.1) |
| University degree | 462 (11.8) | 187 (9.9) | 275 (13.6) |
| **Employment status** | | | |
| Employed | 2404 (61.6) | 1247 (66.3) | 1157 (57.2) |
| Unemployed | 226 (5.8) | 126 (6.7) | 100 (4.9) |
| Not in a labor force | 1273 (32.6) | 508 (27.0) | 765 (37.8) |
| **Self-assessment of material status** | | | |
| Very good/good | 1757 (45.2) | 899 (47.8) | 858 (42.7) |
| Average | 1797 (46.2) | 841 (44.8) | 956 (47.6) |
| Poor/very poor | 333 (8.6) | 139 (7.4) | 194 (9.7) |
| **Smoking habit** | | | |
| Smoker | 1374 (35.2) | 742 (39.4) | 632 (31.3) |
| Former smoker | 548 (14.0) | 323 (17.2) | 225 (11.1) |
| Non smoker | 1979 (50.7) | 816 (43.4) | 1163 (57.6) |
| **Physical inactivity** | | | |
| Yes | 1210 (31.3) | 543 (29.2) | 667 (33.3) |
| No | 2654 (68.7) | 1317 (70.8) | 1337 (66.7) |
| **Alcohol use** | | | |
| Yes | 1736 (45.8) | 1115 (61.7) | 621 (31.3) |
| No | 2055 (54.2) | 692 (38.3) | 1363 (68.7) |

**Table 2. Prevalence of binge drinking by sociodemographic and health-related lifestyle characteristics of respondents in AP Vojvodina, 2024.**

| Sociodemographic and health-related lifestyle characteristics | Binge drinking 2024 | | | | |
|---|---|---|---|---|---|
| | Total | N | % | 95%CI | *p value* |
| **Gender** | | | | | |
| Male | 1601 | 502 | 31.4 | 29.1-33.7 | <0.001 |
| Female | 1855 | 166 | 8.9 | 7.7−10.0 | |
| **Place of settlement** | | | | | |
| Urban | 2079 | 388 | 18.7 | 17.0-20.4 | 0.223 |
| Rural | 1377 | 280 | 20.3 | 18.2-22.6 | |
| **Age groups [years]** | | | | | |
| 18-34 | 815 | 213 | 26.1 | 23.1-29.3 | <0.001 |
| 35-49 | 877 | 210 | 23.9 | 21.2-26.9 | |
| 50-64 | 924 | 177 | 19.2 | 16.1-21.8 | |
| 65+ | 839 | 68 | 8.1 | 6.3-10.2 | |
| **Marital status** | | | | | |
| Married /Living with a partner | 2067 | 386 | 18.7 | 17.0-20.4 | <0.001 |
| Divorced /Separated | 314 | 69 | 22.0 | 17.5-27.0 | |
| Widow/widower | 399 | 25 | 6.3 | 4.1-9.1 | |
| Never married | 670 | 185 | 27.6 | 24.3-31.2 | |
| **Education level** | | | | | |
| University degree | 992 | 194 | 19.6 | 17.1-22.2 | 0.004 |
| Secondary school | 2049 | 419 | 20.4 | 18.7-22.3 | |
| Incomplete primary school/primary school | 413 | 55 | 13.3 | 10.2-17.0 | |
| **Employment status** | | | | | |
| Employed | 2101 | 500 | 23.8 | 22.0-25.7 | <0.001 |
| Unemployed | 192 | 43 | 22.4 | 6.7-29.0 | |
| Not in a labor force | 1158 | 123 | 10.6 | 8.9-12.5 | |
| **Self-assessment of material status** | | | | | |
| Very good/good | 1539 | 314 | 20.4 | 18.4-22.5 | 0.271 |
| Average | 1598 | 290 | 18.1 | 16.3-20.1 | |
| Poor/very poor | 298 | 59 | 19.8 | 15.4-24.8 | |
| **Smoking habit** | | | | | |
| Smoker | 1178 | 366 | 31.1 | 28.4-33.8 | <0.001 |
| Former smoker | 485 | 79 | 16.3 | 13.1-19.9 | |
| Non smoker | 1790 | 223 | 12.5 | 11.0-14.5 | |
| **Physical inactivity** | | | | | |
| Yes | 1057 | 154 | 14.6 | 12.5-16.8 | <0.001 |
| No | 2372 | 507 | 21.4 | 19.7-23.1 | |
| **Total** | **3456** | **668** | **19.3** | **18.0-20.7** | |

both genders. Regarding physical inactivity, male respondents who were not physically inactive had higher odds of being binge drinkers (OR=1.7; 95% CI = 1.3–2.2; p < 0.001). Lower odds of binge drinking were observed among unemployed females compared to those who were employed or self-employed females (OR=0.4; 95% CI = 0.2–1.0; p = 0.036) (Table 3).

**Table 3. Binge drinking in AP Vojvodina according to the sociodemographic and health-related lifestyle factors and multivariable binary logistic regression (gender-stratified odds ratios), 2024.**

| Predictor variables | Total | | Male | | Female | |
|---|---|---|---|---|---|---|
| | OR (95%CI) | p | OR (95%CI) | p | OR (95%CI) | p |
| **Gender** | | | | | | |
| Male | 4.3 (3.5-5.2) | **<0.000** | | | | |
| Female | Ref. | | | | | |
| **Age** | | | | | | |
| 18-34 | 2.6 (1.6-4.0) | **<0.000** | 2.5 (1.4-4.3) | **0.002** | 2.9 (1.3-6.4) | **0.009** |
| 35-49 | 2.2 (1.4-3.5) | **<0.000** | 2.0 (1.2-3.6) | **0.012** | 2.5 (1.1-5.5) | **0.018** |
| 50-64 | 1.7 (1.1-2.6) | **0.015** | 1.7 (1.0-2.9) | **0.042** | 1.5 (0.7-3.2) | 0.237 |
| 65+ | Ref. | | Ref. | | Ref. | |
| **Place of settlement** | | | | | | |
| Urban | Ref. | | Ref. | – | Ref | – |
| Rural | 1.1 (0.9-1.3) | 0.271 | 1.1 (0.8-1.4) | 0.461 | 1.2 (0.8-1.7) | 0.389 |
| **Marital status** | | | | | | |
| Married/living with a partner | Ref. | | Ref. | – | Ref. | – |
| Divorced/separated | 1.2 (0.9-1.7) | 0.153 | 1.1 (0.7-1.7) | 0.561 | 1.4 (0.9-2.5) | 0.135 |
| Widow/widower | 0.8 (0.4-1.3) | 0.342 | 0.8 (0.4-1.7) | 0.559 | 0.8 (0.4-1.7) | 0.616 |
| Never married | 1.1 (0.9-1.5) | 0.232 | 0.9 (0.7-1.3) | 0.612 | 2.1 (1.4-3.5) | **0.002** |
| **Education level** | | | | | | |
| Incomplete primary school/primary school | Ref. | | Ref. | – | Ref. | – |
| Secondary school | 0.9 (0.6-1.4) | 0.786 | 0.9 (0.6-1.4) | 0.804 | 0.9 (0.5-1.8) | 0.820 |
| University degree | 0.9 (0.6-1.4) | 0.812 | 0.9 (0.5-1.5) | 0.705 | 1.0 (0.5-2.1) | 0.995 |
| **Employment status** | | | | | | |
| Employed or self-employed | Ref. | | Ref. | – | Ref. | – |
| Unemployed | 0.7 (0.5-1.2) | 0.223 | 1.0 (0.6-1.7) | 0.807 | 0.4 (0.2-1.0) | **0.036** |
| Not in a labor force | 0.8 (0.6-1.1) | 0.196 | 0.7 (0.5-1.1) | 0.159 | 0.9 (0.6-1.6) | 0.730 |
| **Self-assessment of material status** | | | | | | |
| Very good or good | Ref. | – | Ref. | – | Ref. | – |
| Average | 0.9 (0.7-1.1) | 0.580 | 0.9 (0.7-1.1) | 0.471 | 1.0 (0.7-1.5) | 0.853 |
| Poor or very poor | 1.3 (0.9-2.0) | 0.103 | 1.0 (0.6-1.6) | 0.977 | 2.1 (1.2-3.9) | **0.014** |
| **Smoking habit** | | | | | | |
| Smoker | 2.8 (2.3-3.5) | **<0.001** | 2.7 (2.1-3.5) | **<0.001** | 3.0 (2.1-4.4) | **<0.001** |
| Former smoker | 1.3 (0.9-1.8) | 0.074 | 1.1 (0.7-1.6) | 0.524 | 1.8 (1.1-3.3) | **0.033** |
| Non smoker | Ref. | | Ref. | – | Ref. | – |
| **Physical inactivity** | | | | | | |
| Yes | Ref. | **0.001** | Ref. | **<0.001** | Ref. | 0.554 |
| No | 1.4 (1.2-1.8) | | 1.7 (1.3-2.2) | | 1.1 (0.8-1.6) | |

OR – Odds Ratio; 95% CI – 95% Confidence Interval; Ref – Reference value.

## Discussion

The findings of this study present the data from the new behavioral factors surveillance system on the subnational level, in APV, Serbia ("SBRF-NCD-V"). In our work, we wanted to identify sociodemographic and health-related lifestyle factors that are important covariates of binge drinking. The findings of this study align with drinking rates previously reported in the SNHS conducted in 2019, where almost 50% of the population drank alcohol [21]. According to the results of the

surveillance carried out in Italy that applied the same methodology, 57.9% of respondents consumed alcohol in the 30 days before the survey [28], while according to the latest available results of the BRFSS survey in the US, that percentage was 53.4% [29]. The determined prevalence of binge drinking in 2024 in Vojvodina was 19.3%. These results showed a higher prevalence of binge drinking in comparison with SNHS, where at least once during the month, in the last 12 months, 10.9% of the population of Serbia drank more than six alcoholic drinks on one occasion, 18.3% of men and 4.5% of women [21]. In neighboring countries, using the same methodology as SNHS, that percentage is somewhat higher, reaching 12.6% in the Republic of Croatia [30] and 20.3% in the Republic of Slovenia [31].

Another study in Serbia has revealed that the prevalence of binge drinking in 2014, was 28.4%, and the highest rate was in APV, where every third person was a binge drinker [32]. The different results in our research can be explained by the methodology differences (mostly because of differences in study design and criteria cut-off, such as age limit or observed drinking period). In the literature, there are different but similar indicators for calculating the prevalence of binge drinking. For example, the World Health Organisation (WHO) defines binge drinking as consuming six or more standard drinks on at least one occasion in the last three months [33], and National Institute on Alcoholism and Alcohol Abuse (NIAAA), defines binge drinking as drinking 4 drinks for women and 5 drinks for men in about 2 hours [7]. This discrepancy may also be attributed to differences in sampling methodology, as our sample is based on beneficiaries of compulsory health insurance across all PHCs in APV. The prevalence of binge drinking in our results compared to research that applied the same methodology of drinking patterns is higher than in the United States (US) where 17% of the population were binge drinkers, and in Italy, where 9.6% population was binge drinkers [28,29], but less than in Peru (22.4%) [18].

This research shows that the statistically more binge drinkers in 2024 were among men, younger people, those who have never been married, those with a secondary level of education, those who are employed, smokers, and those who were not physically inactive. Across models, the strongest predictors of binge drinking were male gender and current smoking. Males had several times higher odds of binge drinking, which is consistent with other research [18,32,34–37]. Biological as well as cultural differences are among the main drivers of these differences that need to be further studied [38]. The highest frequency of binge drinking has been found among younger population groups; the odds for binge drinking decrease with age, where adolescents and young adults are the population group at the highest risk of this behaviour, which is consistent with results from Peru, Italy, Spain, and Ontario, Canada [18,28,39,40]. In our research, never-married females had almost 2 times higher chances of binge drinking, compared to married females. Our study confirmed findings that single persons have higher odds of binge drinking [32], but in our research, like in Peru, that is only in females [18]. Research conducted among adults aged 50 and over in 28 European countries and Israel showed that men not living with a spouse, compared to married men, had a higher likelihood of hazardous alcohol use, while in women, widowhood was associated with a lower likelihood of this behavior [36].

The inconsistencies of these results with our research could be explained due to the methodology differences, because in our study, we combined a group of divorced individuals with those who were separated, and a group of married individuals with those living with a partner.

In our study, the higher prevalence of binge drinking was in rural than in urban settlements, although the association was not significant. Similarly, another study conducted in Serbia found no significant association between binge drinking and the type of settlement [32]. In contrast, studies conducted in Ontario, Canada, and Spain revealed that binge drinking was associated with living in rural areas [39,40], while research conducted in several European countries found that living in non-rural residences increased the likelihood of at-risk drinking (Norway, Denmark, and Belgium) [37]. Rates of alcohol use and alcohol use disorder vary with geographic location. Research on risks for alcohol use between rural and urban settings is complicated by the varied systems used to classify locations [20], which may explain differences between our results and those in the literature. Studies comparing the prevalence of heavy or binge drinking based on a dichotomous urban/rural classification have mixed findings when compared with those using more detailed urban-to-rural categories [20].

Regarding education, our findings suggest that the highest percentage of the population who are binge drinkers was among those with secondary education, but in models, educational status wasn't a predictor of binge drinking, which is consistent with research conducted in Serbia in 2014 [32]. However, research conducted in Ontario, Canada, has shown that lower educational attainment was associated with binge drinking [39], while in other countries, higher educational attainment was significantly associated with increased risks of binge or other at-risk drinking compared with formal education [18,37]. A recent study suggested that higher educational attainment is associated with a higher frequency of alcohol use but a lower risk for alcohol dependence [41]. It should be acknowledged that educational classifications may vary between countries, which can explain differences in results.

In Ontario, Canada, unemployed people were more likely to binge drink [39], while in our research, results show that unemployed women had less chance of binge drinking than employed women. That can be due to insufficient income. *Collins* revealed that although groups with greater socioeconomic advantages had similar or greater levels of alcohol consumption than those with fewer advantages, the groups with fewer socioeconomic advantages were at greater risk for alcohol-related problems [12], which can explain why women with poor material status in our results had a higher chance of binge drinking than women with good material status. Research conducted in Serbia in 2014 found no association between binge drinking and self-assessment of socioeconomic status [32], while in Peru, higher income was significantly associated with binge drinking, especially in women [18]. Other research found that binge-drinking prevalence was highest among those with the highest income, but frequency and intensity were highest among those with the lowest income [42]. These differences may be partly explained by differences in the methods used to assess socioeconomic status.

In our research, male respondents who were not physically inactive had higher odds of being binge drinkers. Individuals who drink heavily may engage in frequent physical exercise to compensate for the extra calories gained through drinking or to counterbalance the negative health effects of drinking [43]. Another study found that participants engaging in strengthening and team sports were more likely to binge drink more frequently in the past 30 days than inactive participants [44]. A lifespan study showed that people drank more than usual on the same days that they engaged in more PA than usual [45]. This finding provides insights that alcohol may be used as a reward for being physically active, and, on the other hand, people may be exercising to help offset the caloric intake associated with alcohol use [44]. In our study, we did not analyze the intensity and level of physical activity, which may represent a methodological limitation. In a study conducted on a student population, it was found that among men, strength training is associated with a higher likelihood of binge drinking, whereas this association was not confirmed in women, however, among women, aerobic physical activity was associated with binge drinking. These findings suggest that different forms of physical activity may have distinct patterns of association with risky behaviors [46] A study by Liu et al. also found that this association depends on the type of physical activity—specifically, occupational and transportation-related physical activity are often positively associated with alcohol consumption, while individuals who engage in moderate to vigorous recreational physical activity have a lower likelihood of binge drinking [47].

According to the results of the Behavioral Risk Factor Surveillance System conducted in the United States, the prevalence of binge drinking among smokers is nearly double that of non-smokers [6]. In our study, smokers had almost 3 times higher odds of binge drinking. That follows many results [32,34,37]. The previous study has shown a bidirectional relationship between alcohol and tobacco consumption, where one behavior increases the chance of the other [48]. Female smokers had a higher likelihood of binge drinking compared to males, which follows other research [18,36,49]. Also, women who were former smokers had a higher chance of binge drinking.

## Limitations of the study

The prevalence of alcohol consumption estimated in this study may not accurately represent the prevalence in the overall population because the research is focused on primary healthcare users, and individuals who utilize health services may differ from the general population in factors such as age, gender, and morbidity. We think that these differences are

minimal, given that in Serbia, the rate of compulsory insurance is very high (98%) [23,24], and considering that Serbia's health system is characterized by universal health coverage, ensuring that more than 80% of the population is enrolled with general practitioners and has access to preventive health services. Smoking was included as a predictor of drinking, even though there may be a bidirectional relationship between smoking and drinking. A limitation of this study could also be that alcohol consumption is underreported by giving socially desirable answers, given the sensitivity of the topic. One of the limitations could be that causal relationships can not be determined due to a cross-sectional study design.

## Conclusion

The findings of this study presented the second-year data from the newly developed behavioural risk-factors surveillance system on the sub-national level, in the Autonomous Province of Vojvodina, Serbia ("SBRF-NCD-V"), showing that binge drinking occurs in about one-fifth of adults. The sociodemographic factors in the model vary by gender: males, younger, never-married females, females with poor material status, smokers, females who were former smokers, and physically active males are more likely to binge drink, while unemployed women are less likely. To our knowledge, this is the first study that reports the differential impact of factors associated with binge drinking according to gender in Vojvodina.

Future studies should explore social influences, while current findings highlight the need for monitoring and examination of the influencing factors of binge drinking. Identifying socio-demographic factors may inform clinical intervention in primary care settings to screen and support patients who are at risk for or engaged in risky alcohol use.

## Acknowledgments

The authors are grateful to the people who agreed to participate in the research, to health professionals in 44 Primary Health Care Centers for their help in recruiting and interviewing respondents, and to the whole team, which includes interviewers, supervisors, and a computer engineer, research scientists from IPHV and Faculty of Medicine of the University of Novi Sad.

## Author contributions

**Conceptualization:** Tanja Tomašević, Ivana Radić, Vesna Mijatović Jovanović.

**Data curation:** Tanja Tomašević, Ivana Radić, Snežana Ukropina.

**Formal analysis:** Tanja Tomašević, Ivana Radić, Dragana Milijašević, Sonja Čanković.

**Funding acquisition:** Vladimir Petrović.

**Investigation:** Tanja Tomašević, Ivana Radić, Vesna Mijatović Jovanović, Snežana Ukropina, Sonja Čanković, Radmila Petrović.

**Methodology:** Tanja Tomašević, Ivana Radić, Vesna Mijatović Jovanović, Snežana Ukropina, Dragana Milijašević, Sonja Čanković, Vladimir Petrović.

**Project administration:** Tanja Tomašević, Ivana Radić, Vesna Mijatović Jovanović, Snežana Ukropina.

**Resources:** Tanja Tomašević, Ivana Radić, Sonja Čanković, Vladimir Petrović.

**Software:** Tanja Tomašević, Ivana Radić, Dragana Milijašević.

**Supervision:** Vesna Mijatović Jovanović, Snežana Ukropina, Vladimir Petrović.

**Validation:** Tanja Tomašević, Vesna Mijatović Jovanović, Radmila Petrović.

**Writing – original draft:** Tanja Tomašević, Ivana Radić, Vesna Mijatović Jovanović.

**Writing – review & editing:** Tanja Tomašević, Ivana Radić, Vesna Mijatović Jovanović, Snežana Ukropina, Sonja Čanković, Vladimir Petrović.

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
