## [Decision Letter · Decision Letter 0]

1 Dec 2025

PONE-D-25-45788Predictors of binge drinking among adults in Autonomous Province of Vojvodina, Serbia: data from behavioral risk-factors surveillance systemPLOS ONE

Dear Dr. Radić,

Thank you for submitting your manuscript to PLOS ONE. After careful consideration, we feel that it has merit but does not fully meet PLOS ONE’s publication criteria as it currently stands. Therefore, we invite you to submit a revised version of the manuscript that addresses the points raised during the review process.

We look forward to receiving your revised manuscript.

Kind regards,

Vincenzo De Luca

Academic Editor

PLOS ONE

Journal Requirements:

3. Please note that funding information should not appear in the Acknowledgments section or other areas of your manuscript. We will only publish funding information present in the Funding Statement section of the online submission form. Please remove any funding-related text from the manuscript.

5. For studies involving third-party data, we encourage authors to share any data specific to their analyses that they can legally distribute. PLOS recognizes, however, that authors may be using third-party data they do not have the rights to share. When third-party data cannot be publicly shared, authors must provide all information necessary for interested researchers to apply to gain access to the data. (https://journals.plos.org/plosone/s/data-availability#loc-acceptable-data-access-restrictions)

7. Please amend your manuscript to include a reference list. References must be placed at the end of the manuscript and numbered in the order that they appear in the text.

For more information on the formatting of references, please visit the author guidelines at: http://journals.plos.org/plosone/s/submission-guidelines#loc-reference-style

Reviewers' comments:

Reviewer's Responses to Questions

**Comments to the Author**

1. Is the manuscript technically sound, and do the data support the conclusions?

Reviewer #1: Yes

Reviewer #2: Yes

Reviewer #3: Partly

Reviewer #4: Partly

2. Has the statistical analysis been performed appropriately and rigorously? 

Reviewer #1: Yes

Reviewer #2: Yes

Reviewer #3: No

Reviewer #4: Yes

3. Have the authors made all data underlying the findings in their manuscript fully available?

Reviewer #1: Yes

Reviewer #2: Yes

Reviewer #3: No

Reviewer #4: No

4. Is the manuscript presented in an intelligible fashion and written in standard English?

Reviewer #1: Yes

Reviewer #2: Yes

Reviewer #3: No

Reviewer #4: No

5. Review Comments to the Author

Reviewer #1: All comments are uploaded as a word document. Here is a short summary:

Overall, this is a well-conceived and relevant study addressing an important public health issue. The manuscript is methodologically sound and provides valuable insights into binge drinking patterns in Vojvodina. To further strengthen the manuscript, improvements are recommended in language flow, section transitions, and methodological clarity. Enhancing consistency in academic language, tightening redundancies, and expanding contextual discussion will significantly improve the manuscript’s readability and impact.

Reviewer #2: This is an interesting paper exploring some of the demographic associations with binge drinking. The results are fairly consistent with what has been observed in other countries, though with some interesting differences. The authors focus just on some demographic features as if these should be consistent across cultures. But what about social and cultural factors that might influence drinking behaviour and account for differences in observed rates of binge or other drinking? Also, what about the impact of subcultures on drinking habits? Sports oriented subcultures may have different norms than academically inclined, or religiously oriented subcultures. These may cross the usual demographic boundaries.

I wondered why alcohol use in a general sense was included as a demographic (explanatory factor). Would it not have been useful to know to what degree the demographic factors identified with binge drinking hold for drinking in general? Do we really know how specific these factors are to binge drinking? Could they be non-specific?

Overall, the paper is well written. I found a few typographical errors that should be corrected.

In the introduction, line 51, it is better to say 4 or more drinks for women, 5 or more drinks for men.

Line 94 is a bit ambiguous – better to say adults who used healthcare in one or more of 44 healthcare centers

In the definition of binge drinker, they describe two conditions – binging or non-consumption. Does that mean that they eliminated from the study any non-binge but still drinkers? In that case, what do they know about how their risk variables affect non-binge drinking patterns? Might they also be risk factors for drinking generally? How do we know that they are specific to binge drinking?

Oh, I see alcohol use is an explanatory variable. Still, if it an explanatory variable, how do we see effect on it?

Lines 221, 224, does ‘twice a time’ simply mean ‘two times’ or ‘twice’?

Line 258 ‘criteriums’ should be ‘criteria’

Reviewer #3: Summary: Tomašević et al. investigated risk factors for binge drinking in adult population of Autonomous Province of Vojvodina, Serbia. They included demographics and health behaviors as potential predictors, and the analysis was stratified by sex. Their findings suggested that younger age and current smokers were associated with increased likelihood of binge drinking in both sexes. Other significant variables were never married, low socioeconomic status, and former smokers in females, and physically inactive males.

The comments below are grouped by topics/themes.

1. Need further clarification:

a. Was the BRFSS instrument translated to the primary language(s) spoken in APV prior to administration? If yes, please include this information.

b. Please elaborate on what the BRFSS instrument consists of (e.g., number of items, score range).

c. On the sample size section (line 116), it is stated that “persons with mental impairment” was one of the exclusion criteria. Does this mean any mental health conditions or serious mental illness?

d. On the outcome variable section (line 147), binge drinkers were coded as 1, and non-consumptions were coded as 0. What about those who consumed alcohol in the past 30 days but did not binge? Were they excluded from the study?

2. Statistical analysis:

a. Please report the number and percentage of incomplete/missing/excluded data and the type of missing data, especially if the rate of missingness is substantial.

b. Please check for potential multicollinearity between employment status and socioeconomic status. If they are highly correlated, then consider dropping one of them in the analysis.

c. On the statistical analysis section (line 171), it is stated that p <0.05 was statistically significant. Since there are two separate models (males and females), please consider controlling the rate of false positives (e.g., Benjamini-Hochberg, Bonferroni) or providing reasons to keep the choice of p <0.05.

d. Need to report additional metric, such as goodness-of-fit and/or model performance/discrimination

e. On the abstract (line 27), former smokers in females were a significant predictor. However, the 95% confidence interval includes 1.0. The authors might consider reporting more precise CI values (e.g., two to three decimal places) to avoid confusion.

3. Writing:

a. The authors should revise the language to improve readability.

b. Please check for punctuation.

c. The authors should consider using either ‘sex’ or ‘gender’ consistently throughout the paper.

Reviewer #4: The manuscript addresses an important public health issue and makes valuable use of a large, population-based dataset collected in Vojvodina, Serbia. To further improve the clarity and accuracy of the manuscript, I recommend the following revisions:

I recommend that the authors conduct a thorough language edit of the manuscript to address grammatical errors, inconsistencies, and issues with clarity. Please ensure consistent formatting throughout, including uniform font type and size, and consistent use of punctuation. Additionally, several acronyms are either defined after their first use or not defined at all. Please review and correct all acronym usage.

The term material status appears to be used interchangeably with socioeconomic status throughout the manuscript. However, these concepts are not equivalent. From the Methods section, the authors describe material status as the variable used, yet in other parts of the manuscript the term socioeconomic status is used instead. Material status generally reflects material living conditions or deprivation, while socioeconomic status encompasses broader dimensions such as income, education, and occupation, these terms should not be treated as interchangeable. If the authors intended material status to serve as an indicator of socioeconomic status, then explicit justification of this study design choice and a clear description of how material status was explained and asked during the data collection phase with participants should be provided. If this is not the case, the two terms should not be used interchangeably, as they represent different constructs. Please clarify which construct was actually measured/collected in the study. Please ensure consistent terminology across all sections, and revise to accurately reflect the study design/results.

In the Discussion section, the sentence “which is consistent with results from Peru, Italy, Spain, and Ontario (18, 27, 36, 37)” treats Ontario as if it were a country. Please double check the geographic context (e.g., province in Canada) for clarity. This appears more than once throughout the Discussion section.

Finally, in accordance with the PLOS Data Policy, is the data collected in this study publicly available, either as supplementary material or in an online repository? Please provide the appropriate access information if possible.

6. PLOS authors have the option to publish the peer review history of their article (what does this mean?). If published, this will include your full peer review and any attached files.

Reviewer #1: No

Reviewer #2: No

Reviewer #3: **Yes:** Yumiko Wiranto

Reviewer #4: **Yes:** Julie Tian

---

## [Author Response · Author response to Decision Letter 1]

27 Apr 2026

Reviewer #1: All comments are uploaded as a word document. Here is a short summary:

Overall, this is a well-conceived and relevant study addressing an important public health issue. The manuscript is methodologically sound and provides valuable insights into binge drinking patterns in Vojvodina. To further strengthen the manuscript, improvements are recommended in language flow, section transitions, and methodological clarity. Enhancing consistency in academic language, tightening redundancies, and expanding contextual discussion will significantly improve the manuscript's readability and impact.

The authors are very grateful to the Reviewer, not only for essential insights to improve all parts of the manuscript but also for helping to improve the style of the paper.

1. [Page 1, lines 1–10] The title is clear but could be improved by explicitly noting that the study uses population-based surveillance data, which strengthens its methodological framing.

The author's revision: We accepted the suggestion and change title to: „Predictors of binge drinking among adults in Autonomous Province of Vojvodina, Serbia: data from the population-based behavioral risk-factors surveillance system”.

2. [Page 1, lines 14–22] Ensure consistent use of past tense throughout the abstract to maintain scientific formality and grammatical precision.

The author's revision: We made that correction.

3. [Page 1, lines 23–37] The repetition of 'higher likelihoods of binge drinkers' could be condensed into a single, more fluid sentence to enhance readability.

The author's revision: We rephrase that in : 'higher odds of binge drinkers'

4. [Page 1, line 39] Grammar: add the article 'a' in 'which is a public health priority.'

The author's revision: We made that correction.

5. [Page 2, lines 55–60] Clarify the rationale for focusing on Vojvodina, linking to regional health priorities or previous findings to strengthen contextual justification.

We added an explanation : The focus on the Autonomous Province of Vojvodina is justified by its epidemiological and demographic relevance as a region of Serbia. The Serbian National Health Surveys showed that the prevalence of numerous behavioral risk factors for NCDs, including alcohol consumption, was consistently higher in Vojvodina than in other regions of Serbia. On the other hand, no studies have been conducted on predictors of binge drinking to date.

6. [Page 2, lines 72–83] The shift from global to Serbian context is abrupt. Consider adding a bridging sentence that connects international evidence to local relevance.

We added a sentence: „These patterns have been documented internationally and we want to understand how they manifest in specific national contexts, which is essential for developing effective public health responses“.

7. [Page 2, lines 87–95] Separate the discussion of individual versus societal factors to improve conceptual clarity.

The author's revision: Thank you for the suggestion, but we couldn’t separate individual and societal factors because that would change our concept of discussion.

8. [Page 2, line 111] Use 'Autonomous Province of Vojvodina' consistently instead of 'Autonomy Province' for accuracy.

The author's revision: We made that correction.

9. [Page 3, lines 120–126] Move the sentence on study period to the start of the Methods section for logical flow.

The author's revision: We made that correction.

10. [Page 3, lines 129–133] Clarify whether sampling included all adults or only healthcare users; this affects representativeness.

The answer of the authors: The respondents were primary healthcare users of all healthcare services in one across all 44 PHCs in each of 45 municipalities in APV, considering that Serbia's health system is characterized by universal health coverage, ensuring that more than 80% of the population is enrolled with general practitioners and has access to preventive health services, and sampling process was stratified based on Census data regarding gender, age, and settlement type across two phases (districts and municipalities), which helped align the sample structure as closely as possible with more accurate, but less cost-effective, household-based methods.

11. [Page 3, lines 143–149] The detailed description of healthcare services can be shortened; focus on sampling logic.

The author's revision: We corrected and shortened the text.

12. [Page 3, lines 155–162] Clarify the meaning of 'precision of 1.5%'—cite reference or formula used for calculation.

We added an explanation: A precision of 1.5% was selected to ensure a narrow margin of error (±1.5 percentage points) around the estimated prevalence, allowing for a more accurate and reliable representation of the true population value at a 95% confidence level. We added a reference in the manuscript.

13. [Page 4, lines 170–180] The ethics paragraph is comprehensive but overly long. Consider condensing without losing essential information.

The author's revision: We rewrote the “ethics paragraph” to: „ Ethical aspects of surveillance (compliance of protocol, questionnaire, written informed consent of respondents, and methodological instructions for implementation of the survey with participants' rights, ensuring that the study adheres to regulations, ethical guidelines, and regulatory standards) were approved by the Ethics Committee of the Institute of Public Health of Vojvodina (Decision No. 01-796/2-1 dated May 17, 2024). Each respondent signed the Informed Consent for participation in the survey, and it is stored separately from the database. The survey was anonymous and voluntary, with no incentives provided, and the data were kept confidential on the IPHV server“.

14. [Page 4, line 185] Replace 'Data were completed' with 'Data were collected' for accuracy.

The author's revision: We rephrased 'Data were completed' in 'Data were collected'

15. [Page 4, lines 186–190] Add a short note on data security or storage protocols to reinforce data integrity.

We added an explanation: All data are de-identified and contain no personal information. They are stored on the server of the Institute of Public Health of Vojvodina, with database access restricted to a single computer operated by one engineer. After approval from the Institute’s Ethics Committee, a copy of the database is provided to the researcher for data analysis.

16. [Page 5, lines 215–217] Rephrase the BRFSS definition slightly to demonstrate adaptation to local Serbian context.

The author's revision: „Surveillance of Behavioural Risk Factors for Non-Communicable Diseases in Vojvodina”

17. [Page 5, lines 225–228] Clarify whether the 30-day reference period applies uniformly to all variables.

The author's answer: In our work the 30-day reference period applies to health-related and lifestyle risk factor variables, such as alcohol consumption and physical activity.

18. [Page 6, lines 235–239] Briefly justify inclusion of explanatory variables—are they theory-driven or empirically based?

The answer of the authors: The selection of explanatory variables was empirically driven, based on prior studies that identified their association with alcohol consumption patterns.

19. [Page 6, lines 250–253] Mention statistical software earlier and describe how missing data were handled beyond simple exclusion.

The author's answer: We added in a method: Data were analysed using SPSS (Statistical Package for Social Sciences), version 21, applying descriptive and inferential statistics methods.

20. [Page 6, line 255] Use 'categorical data' instead of 'attributive data' for proper statistical terminology.

The author's revision: We made that correction.

21. [Page 7, lines 270–278] Clarify whether data were weighted to represent the population or analyzed as raw sample values.

The author's answer: We didn’t consider weights in analyses because the sample was not probabilistic, but it was based on data on smoking prevalence of 35.5%, confirmed in 2019 NHS, for the final sampling frame, with a precision of 1.5% and an accuracy of 95%. According to the Census data, proportionally for each of 7 districts of APV and then for each of 45 municipalities (two steps of sampling), the number of surveyed individuals according to gender, age, and type of settlement (urban and rural/other) was previously planned (stratification). Each PHC received a separate stratification plan that included gender, settlement, and age for selecting participants in the sample. The survey was conducted over a period of 3.5 months until the stratified data plan was fulfilled.

22. [Page 8, Table 1] Ensure uniform rounding and consistent presentation of decimals across columns.

The author's answer: We changed the comma to a period when reporting age „The average age of the respondents was 49.3 years”.

23. [Page 9, lines 360–368] Standardize the reporting of confidence intervals to one decimal place (e.g., 29.1–33.7).

The author's answer: The authors checked the tables, and we corrected the confidence interval.

24. [Page 10, lines 390–400] Avoid repeating identical odds ratios across genders; consider summarizing concisely.

The author's answer: We made that correction

25. [Page 11, lines 430–440] Include comparison with regional or Eastern European studies to enhance contextual relevance.

The author's answer: We have added a comparison of the prevalence of binge drinking with countries in the region (Croatia and Slovenia).

26. [Page 12, lines 472–478] Long sentences reduce readability; divide complex arguments into shorter, clearer parts.

The author's revision: We made revisions to certain parts of the text.

27. [Page 13, lines 510–520] Acknowledge potential cross-country differences in educational classifications when comparing results.

The author's revision: We added the sentence: "It should be acknowledged that educational classifications may vary between countries, which can explain differences in results."

28. [Page 14, lines 560–570] Discuss possible bidirectional mechanisms between physical activity and alcohol use for more depth.

The author's revision: We completed this section with 2 more citations. „In our study, we did not analyze the intensity and level of physical activity, which may represent a methodological limitation. In a study conducted on a student population, it was found that among men, strength training is associated with a higher likelihood of binge drinking, whereas this association was not confirmed in women; however, among women, aerobic physical activity was associated with binge drinking. These findings suggest that different forms of physical activity may have distinct patterns of association with risky behaviors (46). A study by Liu et al. also found that this association depends on the type of physical activity—specifically, occupational and transportation-related physical activity are often positively associated with alcohol consumption, whereas recreational physical activity is negatively associated with this behavior or individuals who engage in moderate to vigorous recreational physical activity have a lower likelihood of binge drinking (47).

29. [Page 15, lines 605–615] Highlight that causality cannot be inferred due to the cross-sectional study design.

The author's answer: One of the limitations is that causal relationships can not be determined due to a cross-sectional study design. We added the sentence on the limitations of the study

30. [Page 16, lines 640–660] Tighten the conclusion by avoiding repetition and clearly restating key implications for practice and research.

The author's answer: We have tightened the conclusion by removing repetition and clearly emphasizing the key implications.

General Summary Comment

Overall, this is a well-conceived and relevant study addressing an important public health issue. The manuscript is methodologically sound and provides valuable insights into binge drinking patterns in Vojvodina. To further strengthen the manuscript, improvements are recommended in language flow, section transitions, and methodological clarity. Enhancing consistency in academic language, tightening redundancies, and expanding contextual discussion will significantly improve the manuscript’s readability and impact.

Reviewer #2: This is an interesting paper exploring some of the demographic associations with binge drinking. The results are fairly consistent with what has been observed in other countries, though with some interesting differences. The authors focus just on some demographic features as if these should be consistent across cultures. But what about social and cultural factors that might influence drinking behaviour and account for differences in observed rates of binge or other drinking? Also, what about the impact of subcultures on drinking habits? Sports oriented subcultures may have different norms than academically inclined, or religiously oriented subcultures. These may cross the usual demographic boundaries.

”The author's answer: We thank the reviewer for the insightful comments. We agree that social and cultural factors can play an important role in drinking behavior and may explain some of the differences observed across populations. In our study, we focused on demographic variables due to the limitations of the dataset.

I wondered why alcohol use in a general sense was included as a demographic (explanatory factor). Would it not have been useful to know to what degree the demographic factors identified with binge drinking hold for drinking in general? Do we really know how specific these factors are to binge drinking? Could they be non-specific?

The author's answer: Alcohol was not included as an explanatory variable; we aimed to show the prevalence of alcohol consumption. We assumed that demographic factors might differ between general drinking and binge drinking.

Overall, the paper is well written. I found a few typographical errors that should be corrected.

The author's answer: Thank you for your revision. We made a correction

In the introduction, line 51, it is better to say 4 or more drinks for women, 5 or more drinks for men.

The author's revision: Binge drinking is defined as consuming 4 or more drinks for women and 5 or more drinks for men on a single occasion.

Line 94 is a bit ambiguous – better to say adults who used healthcare in one or more of 44 healthcare centers

The author's answer: We thought they used one of the 44 health centers in the territory of Vojvodina, so we accordingly revised the text: The sample size consisted of 3,910 adults aged 18 and over who used healthcare at one of the 44 Primary Healthcare Centers (PHCs).

In the definition of binge drinker, they describe two conditions – binging or non-consumption. Does that mean that they eliminated from the study any non-binge but still drinkers?

The author's answer: No, those who reported that they drink alcohol but did not report having 5 or more drinks for women or 4 or more drinks for men on one occasion were included in category „no binge drinking“. In that case, what do they know about how their risk variables affect non-binge drinking patterns? Might they also be risk factors for drinking generally? How do we know that they are specific to binge drinking?

Oh, I see alcohol use is an explanatory variable. Still, if it an explanatory variable, how do we see effect on it?

The author's answer: Alcohol use was not included in the analysis as an explanatory variable, we corrected that in the description of variables in the Method section. We used this variable only to describe the sample.

Lines 221, 224, does 'twice a time' simply mean 'two times' or 'twice'?

The author's revision: We made that correction.

Line 258 'criteriums' should be 'criteria'

The author's revision: We made that correction.

Reviewer #3: Summary: Tomašević et al. investigated risk factors for binge drinking in adult population of Autonomous Province of Vojvodina, Serbia. They included demographics and health behaviors as potential predictors, and the analysis was stratified by sex. Their findings suggested that younger age and current smokers were associated with increased likelihood of binge drinking in both sexes. Other significant variables were never married, low socioeconomic status, and former smokers in females, and phy

---

## [Decision Letter · Decision Letter 1]

11 May 2026

Predictors of binge drinking among adults in Autonomous Province of Vojvodina, Serbia: data from the population-based behavioral risk-factors surveillance system

PONE-D-25-45788R1

Dear Dr. Radić,

We’re pleased to inform you that your manuscript has been judged scientifically suitable for publication and will be formally accepted for publication once it meets all outstanding technical requirements.

Kind regards,

Vincenzo De Luca

Academic Editor

PLOS One

Additional Editor Comments (optional):

Reviewers' comments:

Reviewer's Responses to Questions

**Comments to the Author**

1. If the authors have adequately addressed your comments raised in a previous round of review and you feel that this manuscript is now acceptable for publication, you may indicate that here to bypass the “Comments to the Author” section, enter your conflict of interest statement in the “Confidential to Editor” section, and submit your "Accept" recommendation.

Reviewer #2: All comments have been addressed

2. Is the manuscript technically sound, and do the data support the conclusions?

Reviewer #2: Yes

3. Has the statistical analysis been performed appropriately and rigorously? 

Reviewer #2: Yes

4. Have the authors made all data underlying the findings in their manuscript fully available?

Reviewer #2: Yes

5. Is the manuscript presented in an intelligible fashion and written in standard English?

Reviewer #2: Yes

6. Review Comments to the Author

Reviewer #2: (No Response)

7. PLOS authors have the option to publish the peer review history of their article (what does this mean?). If published, this will include your full peer review and any attached files.

Reviewer #2: No

---

## [Editor Report · Acceptance letter]

PONE-D-25-45788R1

PLOS One

Dear Dr. Radić,

I'm pleased to inform you that your manuscript has been deemed suitable for publication in PLOS One. Congratulations! Your manuscript is now being handed over to our production team.

Kind regards,

on behalf of

Dr. Vincenzo De Luca

Academic Editor

PLOS One